# Vertically Arranged Zinc Oxide Nanorods as Antireflection Layer for Crystalline Silicon Solar Cell: A Simulation Study of Photovoltaic Properties

**Deb Kumar Shah** [1] [ID], **Devendra KC** [2], **M. Shaheer Akhtar** [1,3,4,*] [ID], **Chong Yeal Kim** [4] and **O-Bong Yang** [1,3,4,*]

1    School of Semiconductor and Chemical Engineering, Jeonbuk National University, Jeonju 54896, Korea; dkshah149@gmail.com
2    Department of Science, Lebesby Oppvekstsenter School, 9740 Lebessby, Norway; devendrakc25@gmail.com
3    Graduate School of Integrated Energy-AI, Jeonbuk National University, Jeonju 54896, Korea
4    New and Renewable Energy Materials Development Center (NewREC), Jeonbuk National University, Jeonbuk 56332, Korea; kimbo@jbnu.ac.kr
*    Correspondence: shaheerakhtar@jbnu.ac.kr (M.S.A.); obyang@jbnu.ac.kr (O.-B.Y.)

**Abstract:** This paper describes the unique antireflection (AR) layer of vertically arranged ZnO nanorods (NRs) on crystalline silicon (c-Si) solar cells and studies the charge transport and photovoltaic properties by simulation. The vertically arranged ZnO NRs were deposited on ZnO-seeded c-Si wafers by a simple low-temperature solution process. The lengths of the ZnO NRs were optimized by changing the reaction times. Highly dense and vertically arranged ZnO NRs were obtained over the c-Si wafer when the reaction time was 5 h. The deposited ZnO NRs on the c-Si wafers exhibited the lowest reflectance of ~7.5% at 838 nm, having a reasonable average reflectance of ~9.5% in the whole wavelength range (400–1000 nm). Using PC1D software, the charge transport and photovoltaic properties of c-Si solar cells were explored by considering the lengths of the ZnO NRs and the reflectance values. The 1.1 μm length of the ZnO NRs and a minimum average reflectance of 9.5% appeared to be the optimum values for achieving the highest power conversion efficiency of 14.88%. The simulation study for the vertically arranged ZnO NRs AR layers clearly reflects that the low-temperature deposited ZnO NRs on c-Si solar cells could pose a greater prospect in the manufacturing of low-cost c-Si solar cells.

**Keywords:** zinc oxide; thin film; silicon solar cells; antireflection layer; power conversion efficiency; PC1D simulation

## 1. Introduction

Recent research in Si solar cell technology is basically focused on reducing the manufacturing cost by means of using low-cost and effective raw materials [1]. Among all solar materials, a coating of a thin layer, called the antireflection (AR) layer, on Si surfaces is applied for reducing the incident reflection and to increase light absorption. In commercial c-Si solar cells, a silicon nitride (SiNx) layer is generally used as an AR material because it shows good attachment with passivated Si surfaces [2]. The expensive Plasma Enhanced Chemical Vapor Deposition (PECVD) method has been applied to deposit SiNx AR layers. The PECVD process is usually faced with several disadvantages, such as controlling and damaging the Si substrate with plasma during the collision of the Si surface and secondary electrons [3,4]. Research communities are uninterruptedly concentrating their efforts to explore cost-effective and ecofriendly photovoltaic (PV) technology [5,6]. Various AR materials having high refractive indexes, such as $Si_3N_4$, a-$SiN_x$, SiO, $SiO_2$, $TiO_2$, $Ta_2O_5$, $MgF_2$, $SiO_2$–$TiO_2$, and ZnS, have

been applied in the fabrication of c-Si solar cells [7–12]. Zinc oxide (ZnO) has a variety of favorable properties, such as high electron mobility, a wide bandgap of 3.37 eV, and strong room temperature luminescence for many electronic and electrical devices [13,14]. ZnO, as a promising AR material, is receiving much attention due to its attractive dielectric properties, good transparency, showing an adjustable refractive index, and excellent capability to grow a textured layering or coating via anisotropic growth [15–17].

In that context, growth of ZnO nanorods (NRs) on Si wafers for light trapping and minimizing the reflectivity of the solar spectrum could be an efficient tool for application in Si solar cells [18]. ZnO NRs show great potential for the optical imprisonment effect, which makes the proper AR materials for high-performance c-Si solar cells [19]. The growth of nanostructures like nanowires, nanorods, nanopyramids, and nanopillars over c-Si wafers can efficiently reduce the reflection by c-Si solar cells [20]. ZnO NR arrays on Si substrates have demonstrated unique optical absorption and excellent trapping of incident light by the optical antenna effect [21]. The ZnO NRs as AR layers considerably induce reflection in a broad wavelength region at different incident angles because continuous transition in the effective refractive index can be regarded across the air-to-wire interface [22–25]. The AR behavior of the ZnO NRs has owned a lot of attention because of its low dielectric constant and appropriate refractive index. Without a proper AR layer, c-Si solar cells face significant energy loss and less absorption of light. Therefore, it has been realized that the deposition of vertically arranged NRs could be the promising means to achieve low-cost solar cells [26–28]. Including other deposition techniques for ZnO NRs, the low-temperature solution method can offer a highly suitable method for controlling NR sizes and densities over Si surfaces at low temperatures.

The absorption of light or multiple reflections can be realized by deploying multiple transparent layers or zigzag structures on the top surface of c-Si solar cells [29]. In our previous work, Jannat, A. et al. prepared SiC-SiO$_2$ nanocomposite AR material, coated by spin coating on Si solar cells, and manifested a reasonable reduction in reflectance to 7.08%. SiC-SiO$_2$ AR-coated Si solar cells presented a power conversion efficiency (PCE) of 16.99%, similar to SiNx AR-coated Si solar cells [30]. Further, Jung, J. et al. successfully deposited the double layer TiO$_2$/Al$_2$O$_3$ anti-reflection coating (ARC) on a Si wafer, which recorded a low reflectance of 4.74% and expressed a higher PCE (~13.95%) than those of single layer TiO$_2$ and Al$_2$O$_3$ ARs [31]. Recently, Shah, D.K. et al. reported on the deposition of a double ZnO/Ag-doped ZnO AR layer over P-type Si solar cells, using a low-cost sol–gel technique via spin coating. It was found that the double ZnO/Ag-ZnO AR layer on c-Si solar cells presented the lowest average reflectance of ~7.13%, as compared with single ZnO and single Ag-doped ZnO AR layers on Si wafers. The fabricated Si solar cell with a double ZnO/Ag-doped ZnO AR layer attained the highest PCE of 9.48%, while the fabricated crystalline Si solar cells with single ZnO and single Ag-doped ZnO AR layers obtained low efficiencies of 8.59% and 8.78%, respectively [4]. Therefore, the reflectivity highly decreased with the growing of rod-like nanostructures on the top surface of the Si substrate, and ZnO NRs can act as an AR layer. It has been proven that there is no need to provide an additional AR layer, which reduces the production cost of solar cells. In this work, the vertically arranged ZnO NRs, as AR layers, have been grown on a c-Si wafer by a low-temperature solution method, and the structural, morphological, optical, and photovoltaic properties were determined. Vertically arranged ZnO NRs on Si wafers suppressed an average reflection of ~9.5% in the wavelength range of 400–1000 nm. Using the length and reflectance of a ZnO NRs AR layer, the photovoltaic parameters were extracted by a simulation study via PC1D software. The fabricated Si nanostructures on the Si solar cell were thoroughly characterized in terms of morphology, structure, crystalline nature, and optical and PV properties.

## 2. Materials and Methods

### 2.1. Deposition of Vertically Arranged ZnO NRs AR on c-Si Wafer

A P-type commercial c-Si wafer ($40 \times 40$ mm$^2$) with a thickness of 120 µm and sheet resistance of 1–3 Ωcm was used as a substrate. The textured Si wafer was thoroughly cleaned by sonication, using acetone and deionized (DI) water for 15 min each. For seed layer deposition, a solution was prepared by mixing 0.9174 g (0.1 M) of zinc acetate hydrate in 50 mL of ethanol. The ZnO seed layer was deposited on the surface of a pre-cleaned Si wafer and a textured Si wafer by spin coating at 2000 rpm for 30 s, and it was annealed in a furnace at 100 °C for 1 h. For the growth of ZnO NRs on the surface of the Si wafer, a solution was prepared by mixing of 0.1 M of zinc nitrate hydrate ($Zn(NO_3)_2$) and 0.1 M hexamethylenetetramine ($C_6H_{12}N_4$) with 50 mL of DI water. The ZnO-seeded Si substrate was dipped vertically in an airtight vessel and kept in an oven at 75 °C for 3, 4, and 5 h. After a specific time, the Si wafer was taken out gently and washed with ethanol and DI water several times. Finally, it was annealed in a furnace at 250 °C for 1 h, as described in Figure 1. For optimization of the morphology of ZnO NRs on Si substrates, both bare and textured Si wafers were used to grow ZnO NRs. The morphologies and optical properties of the grown ZnO NRs were investigated, analyzed, and characterized by a variety of analytical tools like field emission scanning electron microscopy (FESEM), ultraviolet–diffuse reflectance spectroscopy (UV-DRS) spectroscopy, photoluminescence (PL) spectroscopy, Raman analysis, and PC1D simulation tools.

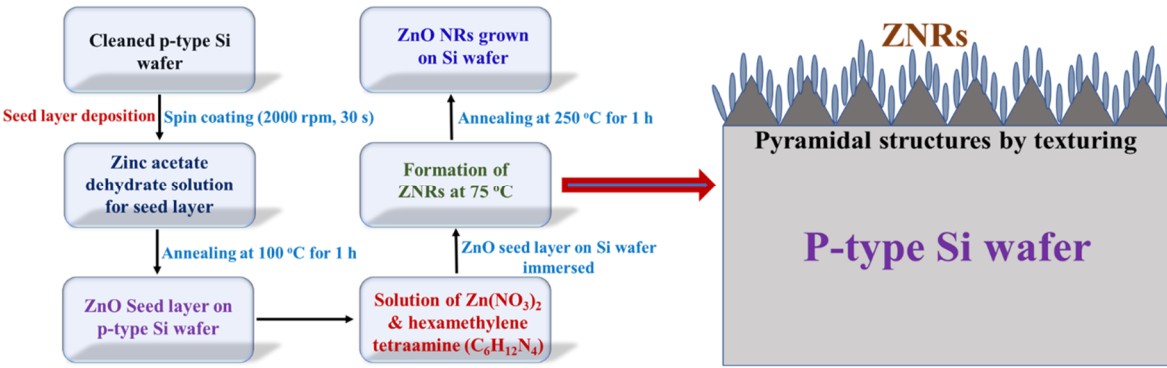

**Figure 1.** Schematic diagram of growth of ZnO nanorods (NRs) on a textured Si wafer.

### 2.2. PC1D Modeling Tool

The numerical modeling software PC1D has been used to simulate the PV properties of c-Si solar cells, and this software was invented by the group at the University of New South Wales [32]. PC1D allows us to simulate the important photovoltaic properties of solar devices fabricated with semiconducting materials by regarding them as one-dimensional (axial symmetry). The PC1D software consists of a large number of libraries and files, including the parameters of a variety of PV technologies like Ge, GIN, GaAs, a-Si, InP, c-Si, and AlGaAs [33,34]. Several key parameters, including active area, thickness of device, band gap, and reflectance, were used to perform the simulation for the photovoltaic parameters of Si solar cells. In this work, the PC1D simulation was performed using the NR lengths and the average reflectance of the ZnO NRs AR layer over the textured c-Si wafer, and the predicted performance and charge transport properties of the solar cells were based on the vertically arranged ZnO NRs AR layer. The details of the simulation parameters are summarized in Table 1. The simulation was performed under AM 1.5 solar radiation and a constant light intensity of 0.1 W/cm$^2$ (one sun) at 300 K. For all PV simulations, the bulk recombination was set to 10 µs, and P-type background doping of the solar cells was set to $1.153 \times 10^{16}$ cm$^{-3}$ and N-type first front diffusion to $2.87 \times 10^{20}$ cm$^{-3}$ peaks.

**Table 1.** Primary parameters used in the PC1D simulation.

| Parameters | Values |
|---|---|
| Device area | 16 cm$^2$ |
| Front surface texture depth | 0–5 μm |
| Front reflectance | 2–4% |
| Thickness of Si solar cell | 120 μm |
| Dielectric constant | 11.9 |
| Energy band gap | 1.124 eV |
| Background doping P-type | $1.513 \times 10^{16}$ cm$^{-3}$ |
| First front diffusion N-type | $2.87 \times 10^{20}$ cm$^{-3}$ |
| Refractive index | 3.58 |
| Excitation mode | Transient |
| Temperature | 25 °C |
| Other parameters | Internal model of PC1D |
| Primary light source | AM 1.5D spectrum |
| Bulk recombination | 10 μs |
| Constant intensity | 0.1 W/cm$^2$ |

### 2.3. Characterizations

The fabricated ZnO NRs were characterized by a variety of analytical, spectroscopic, and PV measurement techniques. For microstructure and surface morphology, a field emission scanning electron microscopy (FESEM, Hitachi 4800, Japan) was used. The confirmation of deposition of ZnO was verified by X-rays diffraction (XRD), photoluminescence (PL) spectroscopy and Raman scattering spectroscopy. The optical properties, such as reflectance of rod nanostructures, were investigated by ultraviolet diffused reflectance spectroscopy and recorded by a Shimadzu MPC-3100 scanning spectrophotometer in the wavelength range of 400–1200 nm. The detailed PV properties were derived by PC1D simulation by using the length of the ZnO NRs and the average reflectance as input parameters.

### 3. Results

To check the uniformity of ZnO nanostructures on the surface of bare as well as textured Si wafers, top and cross-sectional views of the FESEM analysis were observed. The top surface view of ZnO NRs on bare and textured Si wafers presents uniform and well-distributed nanorods over the Si surface, as shown in Figure 2a–h. At the deposition time of 5 h, the highly dense hexagonal faces of the ZnO NRs have grown over the bare Si wafer (Figure 2c). The ZnO NRs were multidirectional on the textured surface, but all NRs were arranged in a vertical manner, as displayed in Figure 2e–h. The top view of the ZnO NRs grown at 5 h reveals that the NRs are well grown in many directions, as they are clearly visible in the cross-sectional view. Figure 3a shows the highly dense NRs over one pyramid of a Si wafer. The cross-sectional view of ZnO NRs grown at 5 h is a well-distributed and multidirectional arrangement of NRs, as shown in Figure 3b.

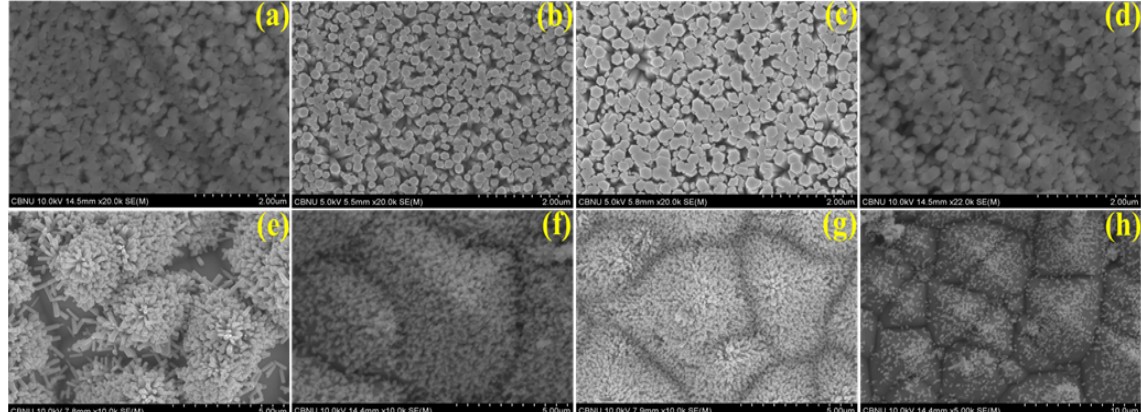

**Figure 2.** Field emission scanning electron microscopy (FESEM) image of the top view of ZnO NRs of treatment times (**a**) 3 h, (**b**) 4 h, (**c**) 5 h, and (**d**) 6 h on bare Si substrate and (**e**) 3 h, (**f**) 4 h, (**g**) 5 h, and (**h**) 6 h on textured Si substrate.

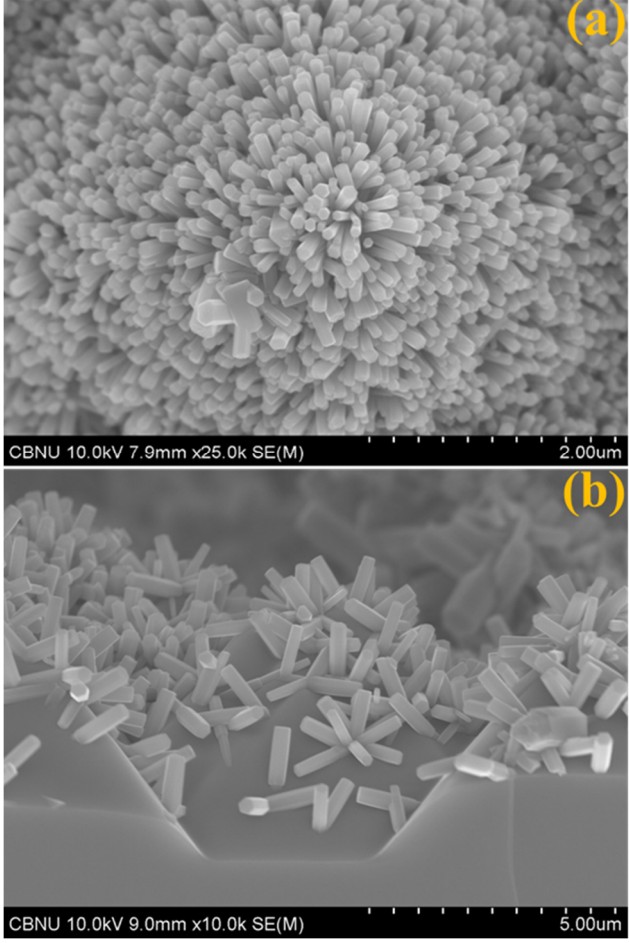

**Figure 3.** FESEM image of (**a**) top and (**b**) cross-sectional view of ZnO NRs for 5 h.

The growth of ZnO NRs on the surface of the textured Si wafer were analyzed to examine the crystalline phases by XRD measurements. XRD patterns of all ZnO NRs AR layers exhibit all the characteristics of the diffraction peaks of ZnO, along with a strong Si phase, which are located at 34.38° (002) and 69.26° (001), respectively, as shown in Figure 4. The observed diffraction peaks confirm the perfect crystal of ZnO, having the hexagonal wurtzite phases of ZnO materials [35] along with the monocrystalline crystalline phase of Si. This examination clearly deduces the growth of single

crystalline ZnO NRs on a Si surface via the low-temperature solution method, followed by annealing at 250 °C for 1 h.

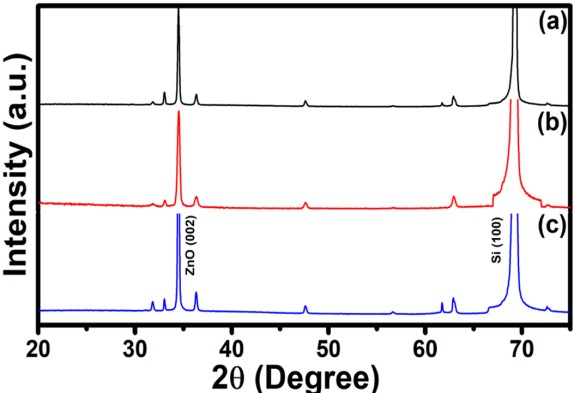

**Figure 4.** XRD Analysis of ZnO NRs on the surface of Si at (**a**) 3 h, (**b**) 4 h, and (**c**) 5 h treatment time.

Photoluminescence (PL) spectroscopy in Figure 5a revealed emission peaks at 393 nm for ZnO NRs at 3 h, 4 h, and 5 h, respectively. The small emission peak at 469 nm belongs to the Si wafer. The PL intensity increased as the reaction time increased, due to an increase in length of the ZnO NRs on the Si substrate. The existence of a prominent PL emission near 400 nm confirms the formation of ZnO on the Si wafer [36]. Moreover, the structural behavior of grown ZnO NRs is further elucidated by the Raman spectroscopy (Figure 5b), which exhibits a sharp Raman band at 437 cm$^{-1}$ with a Si peak. The Raman band at 437 cm$^{-1}$ corresponds to the typical E$_2$ mode, which normally originates from the hexagonal wurtzite phase of ZnO, deducing the creation of ZnO NRs on the Si surface. No other Raman bands were detected, confirming the high purity ZnO crystal without any surface defects.

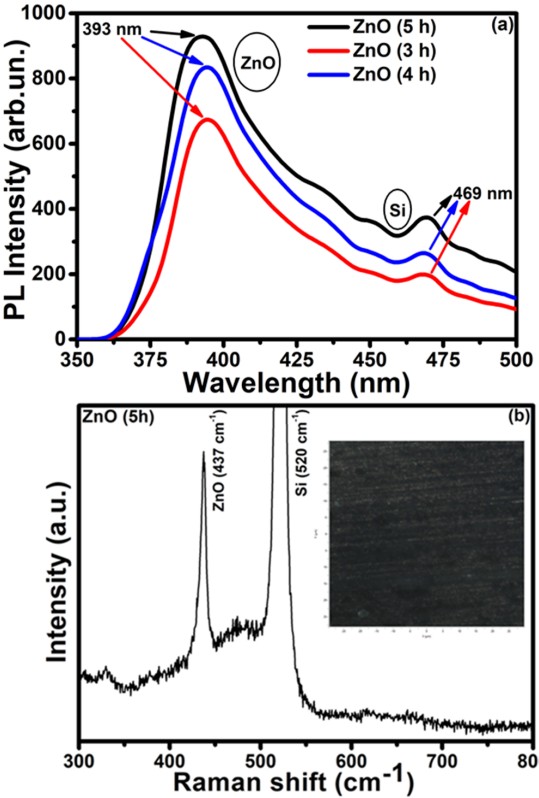

**Figure 5.** (**a**) Photoluminescence and (**b**) Raman spectrum, with an inset image of ZnO NRs on a Si wafer for 5 h of treatment time.

Ultra-violet diffuse reflectance spectroscopy (UV-DRS) has been carried out to evaluate the optical properties of vertically arranged ZnO NRs on Si wafers with reaction times of 3 h, 4 h, 5 h, and 6 h. In Figure 6, the bare Si wafer displays an average reflectance of ~38% in the wavelength range of 400–1000 nm, whereas after texturing the Si wafers, the average reflectance has reduced to 14%. The vertically arranged ZnO NRs on the Si wafer detected an average reflectance of ~12.36% for 3 h, 12.69% for 4 h, 9.58% for 5 h, and 14.95% for 6 h, covering a large wavelength range from 400–1000 nm. As seen in Figure 6, the significant reduction in the reflectance to ~9.58% was recorded in the 5 h reaction time on the Si wafer, which might be due to the generation of multiple reflections of the incident light by the vertically arranged nanorods. The optimized average reflectance of 9.58% at 5 h of treatment time (experimental) is well matched with the average reflectance of 9.79% obtained by the simulation, which is clearly validated in the simulation data as shown in Figure 6.

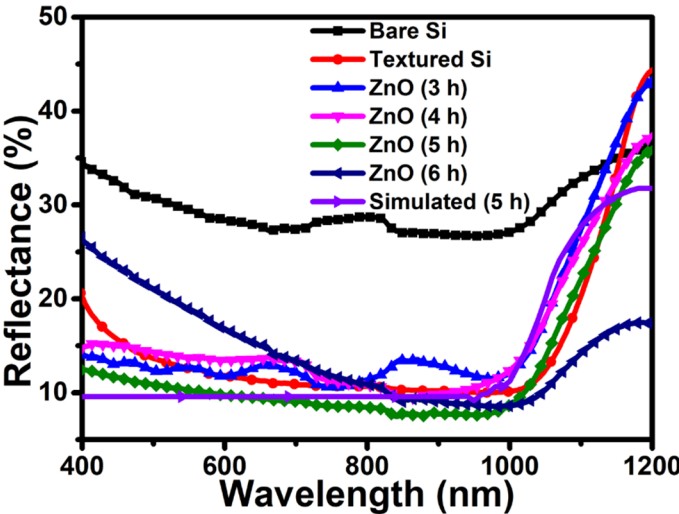

**Figure 6.** UV-Vis spectra in reflectance mode of the bare Si wafer, textured Si wafer, and vertically arranged ZnO NRs on textured Si wafers.

The other optical properties, such as the refractive index (n) and extinction coefficient (k) of ZnO NRs on Si wafers, are crucial to develop efficient Si solar cells, as explored in Figure 7, and are extracted from the reflectance results [37–39]. As compared with other reaction times, the highest refractive index of 1.92 and lowest extinction coefficient are observed by the vertically arranged ZnO NRs on Si wafers with an optimized reaction time of 5 h, as shown in Figure 7a,b. Noticeably, the low k value in the whole visible region represents the weak absorption and large photonic energy of light via the black colored surface of the Si wafer.

Based on the above structural and optical properties of ZnO NRs AR layers, a simulation has been carried out using PC1D simulation tools to investigate the PV properties by simulating the current (I)-voltage (V) and incident photon-to-electron conversion efficiency (IPCE) data. Table 2 summarizes the extracted PV parameters, which were taken to perform the simulation for ZnO NRs AR-based Si solar cells. In this work, the PCE and fill factor (FF) of the solar cells were calculated in accordance with the report's articles [40]. Figure 8a shows the simulated I-V characteristics and P-V curve of the Si solar cell based on ZnO NRs AR. As seen in Table 2, the maximum $I_{sc}$ = 0.49 A and $V_{oc}$ = 0.59 V with a conversion efficiency of 14.88%, observed at a length of 1.1 μm for ZnO NRs and with a minimum reflectance of 9.58%. Solar cells with ZnO NRs 1.1 μm in length show the highest power of ~0.238 W.

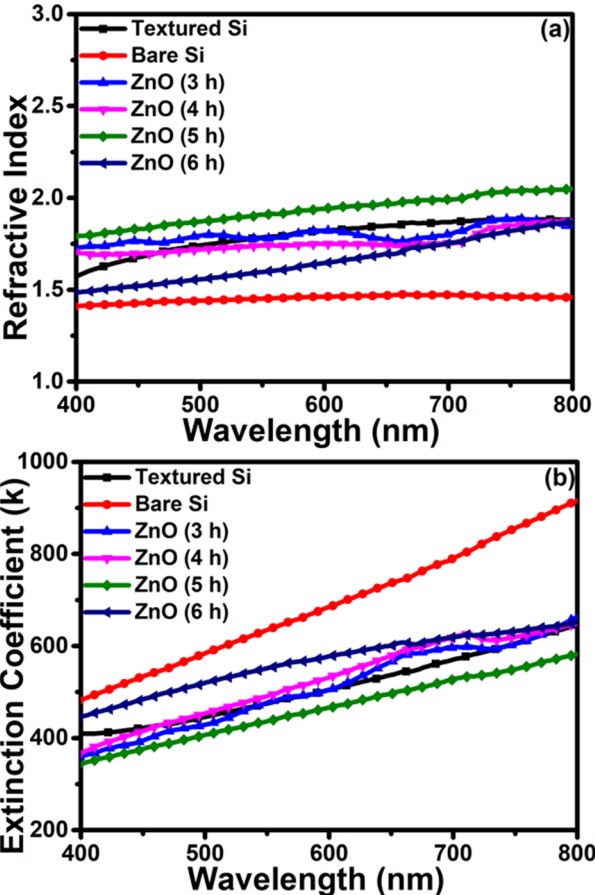

**Figure 7.** (**a**) Refractive index and (**b**) extinction coefficient plots of the bare Si wafer, textured Si wafer, and vertically arranged ZnO NRs on textured Si wafers.

**Table 2.** Summary of optical and PV properties of ZnO NRs AR-based Si solar cells (PC1D simulation).

| Reaction Time (h) | ZnO NRs Length (μm) | Reflectance (%) | Simulated PV Properties | | | | | |
|---|---|---|---|---|---|---|---|---|
| | | | $I_{sc}$ (A) | $V_{oc}$ (V) | $I_{max}$ (A) | $V_{max}$ (V) | Fill Factor | Efficiency (%) |
| 3 | 0.75 | 12.360 | 0.4803 | 0.5889 | 0.455443 | 0.505304 | 81.36 | 14.38 |
| 4 | 1.00 | 12.690 | 0.4796 | 0.5886 | 0.454993 | 0.504741 | 81.35 | 14.35 |
| 5 | 1.10 | 9.580 | 0.4970 | 0.5894 | 0.469686 | 0.507107 | 81.31 | 14.88 |
| 6 | 0.90 | 14.95 | 0.4653 | 0.5870 | 0.441221 | 0.503526 | 81.34 | 13.88 |

A similar trend is noticed in the PV parameters, as obtained in reflectance and other optical results. Interestingly, all the simulated data of solar cells are extracted by putting in the information of the ZnO NRs AR layer, which means vertically arranged NRs are crucial in this system. Furthermore, the plots of internal quantum efficiency (IQE) and external quantum efficiency (EQE) versus wavelength for the ZnO NRs AR-based Si solar cell is shown in Figure 8b. The EQE of a Si solar cell based on ZnO NRs demonstrated over 80% absorption, covering a 400–1200 nm wavelength range, and the integrated photocurrent was consistent with the $I_{sc}$ extracted from the I-V curve. The absorption of blue photons might usually occur at the topmost part of ZnO NRs, where the charge carriers are supposed to move the entire length of the ZnO NRs to reach the p–n junction. Therefore, the enhancement of the PCE of a Si solar cell occurs by controlling the AR behavior of ZnO NRs.

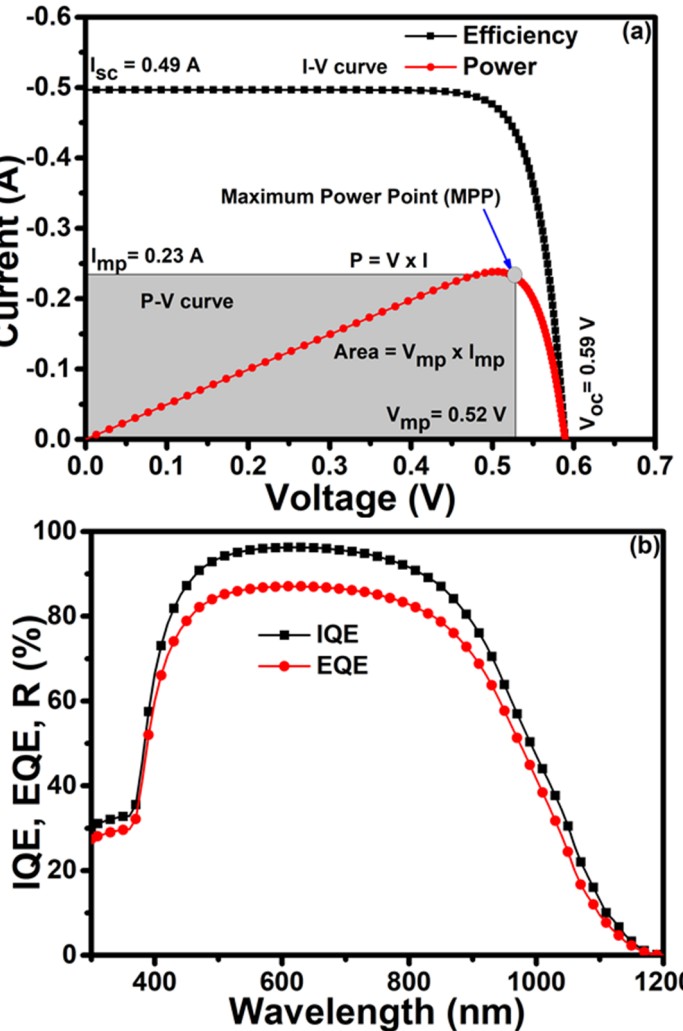

**Figure 8.** (**a**) I-V curve with a power curve and (**b**) incident photon-to-electron conversion efficiency (IPCE) curve of Si solar cells based on ZnO NRs antireflection (AR).

To explain the good PCE and photocurrent of Si solar cells based on ZnO NRs AR, the diffusion coefficient and diffusion length are the critical factors, as they are associated with charge carrier transport upon light illumination [41]. It is known that the shorter diffusion length results in a high recombination rate [41]. Figure 9 depicts the plot of diffusion length and distance from the front of the AR, obtained by PC1D simulation. In all lengths of ZnO NRs AR, the diffusion lengths gradually increased up to 100 μm. Afterward, a saturation in diffusion length was obtained with the increase of distance from the front. This observation clearly deduces that the maximum limit of the diffusion length is 100 μm for all lengths of ZnO NRs, as illustrated in Figure 9. In this calculation, the thicknesses of the emitter regions are 15 μm, 24 μm, 24.78 μm, and 30 μm for 0.75 μm, 0.9 μm, 1.0 μm, and 1.1 μm lengths of ZnO NRs AR, respectively. From Figure 9, the ratios of the diffusion length to the thickness of the emitter are 6.66, 4.16, 4.03, and 3.33 for 0.75 μm, 0.9 μm, 1.0 μm, and 1.1 μm lengths of ZnO NRs, respectively. The lowest ratio value is related to a high penetration of dopant in the emitter. Moreover, the probability of charge collection at the interface of the N and P layers is linked with the diffusion length and size of the device, especially the emitter size [42]. Suppose the thickness of the emitter is lower than the diffusion length, which results in a considerably high charge collection probability. As represented in Figure 9, the high diffusion length obtained by ZnO NRs AR is attributed to a higher probability of charge collection, resulting in the large collection of light-generated charge carriers at the p–n junction and high conduction to the cell. Hence, this simulation study reflects that the

low-temperature solution method is highly feasible for growing vertically arranged ZnO NRs AR to fabricate the Si wafer-based solar cells.

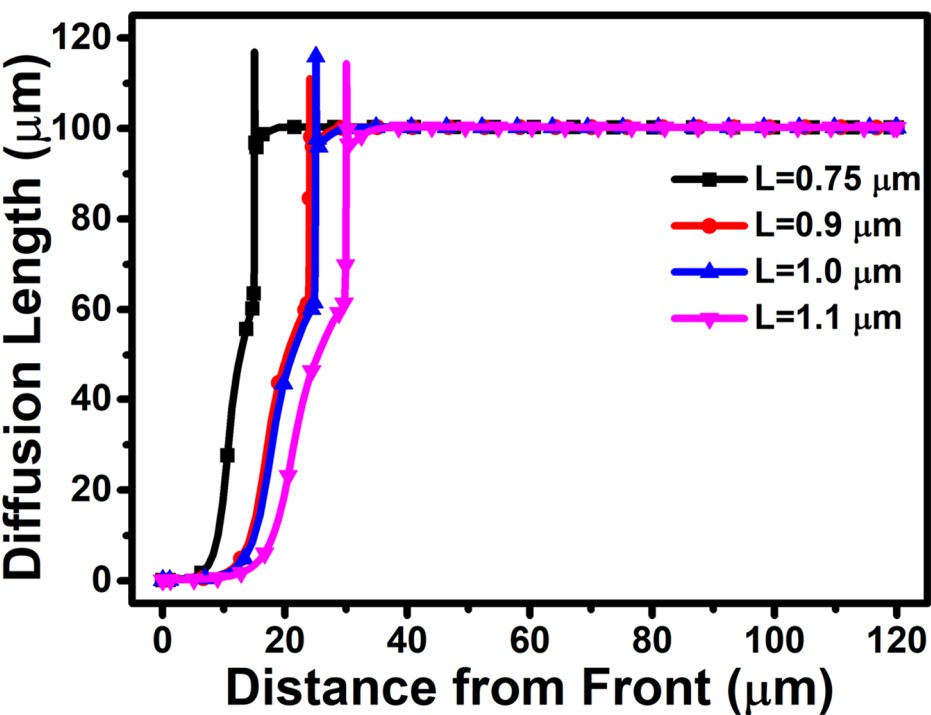

**Figure 9.** Influence of variation in the length of ZnO NRs AR on diffusion length.

## 4. Conclusions

The vertically arranged ZnO NRs were successfully grown on the bare and textured Si wafers by the low-temperature solution method and employed to estimate the PV parameters by PC1D simulation for c-Si solar cell-based ZnO NRs AR. The reaction time has been optimized, in which a 5 h reaction time presented the best surface and highly uniform ZnO NRs on the Si wafer. The grown ZnO NRs exhibited a minimum reflectance of ~7.5% at 838 nm, with a ~9.5% average reflectance in the wavelength range from 400–1000 nm. The length of the ZnO NRs and the average reflectance were applied as input parameters in the PC1D simulation to instigate the PV properties of the solar cell. The optimum conversion efficiency of 14.88% has been achieved in ZnO NRs 1.1 μm in length, with a minimum average reflectance of 9.5%, by using this simulation tool. From this study, it has been confirmed that the reported method (simple hydrothermal method) is an easy, facile method for the preparation of zinc nano rod structure-based Si solar cells, which has greater prospects in developing c-Si solar cells without AR layers at a low cost.

**Author Contributions:** Experiment, investigation, data curation, software, writing, D.K.S.; resources, software, validation, formal analysis, D.K.; writing—review and editing, visualization, M.S.A.; project administration, writing—review and editing, C.Y.K.; funding acquisition, supervision, O.-B.Y. All authors have read and agreed to the published version of the manuscript.

**Funding:** This research received no external funding.

**Acknowledgments:** This work was supported by "Human Resources Program in Energy Technology" of the Institute of Energy Technology Evaluation and Planning (KETEP) and granted financial resources from the Ministry of Trade, Industry & Energy, Republic of Korea (Project No.: 20204010600470). This research was supported by technology development project sponsored by the Korean Government (Project Number: S2644695).

**Conflicts of Interest:** The authors declare no conflict of interest.

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
