# Peer review of "Vertically Arranged Zinc Oxide Nanorods as Antireflection Layer for Crystalline Silicon Solar Cell: A Simulation Study of Photovoltaic Properties"

_applsci, doi:10.3390/app10176062_

Round 1

Reviewer 1 Report

This manuscript showed the antireflection layer of vertically arranged ZnO NRs on c-Si solar cells and studied the charge transport and photovoltaic properties by simulation. The authors optimized the reaction time for best surface and highly uniform ZnO NRs on the Si wafer. The optimum conversion efficiency of 14.88% has been achieved at 1.1 µm of ZnO NRs length with minimum average reflectance 9.5% by using this simulating tool. However, the concept of this manuscript is similar to a previous report (doi:10.1016/j.solmat.2010.01.005). The previous report also used ZnO NRs as antireflection layer on silicon solar cells and showed that the efficiency was improved. When comparing the author's paper with the previous paper published in 2010, I could not find the difference and excellence. Also, there is a lack of explanation in this manuscript.

  1. The authors compared the different reaction time for 3, 4 and 5 hours. Among them, the performance of the simulation for 5 h ZnO NRs AR based Si solar cells shows the highest efficiency. The author is better to compare the 6 h ZnO NRs because the 5 h ZnO NRs lacks credibility to say that it is the optimal reaction time.
  2. In figure 5(a), please explain the reason for the shift of the wavelength and increase of the intensity by changing the reaction time.
  3. Why doesn’t the intensity of the sample (b) 4h show a Si (100) peak?

For these reasons, I do not recommend the publication of this manuscript in Applied Sciences.  

Author Response

Responses to Reviewers#1

Dated: August 20, 2020

Prof. Sadia Ameen

Guest Editor

Applied Sciences

Dear Editor

Thank you for forwarding the reviewer’s comments about our manuscript entitled “Vertically Arranged Zinc oxide Nanorods as Antireflection layer for Crystalline Silicon Solar cell: A Simulation Study of Photovoltaic Properties” with manuscript reference number “applsci-901441” by D.K. Shah et al.

Please find attached herewith the revised manuscript along with the reply sheet to honorable referees. You have kindly given us an opportunity for the revision after doing the needful modifications as per reviewer’s suggestions. Now, we have carefully revised and compensated all the reviewer’s comments in the revised manuscript. The corrected parts are highlighted by yellow background in the revised manuscript.

We hope that we have satisfactorily addressed all the valuable comments of the reviewers.

While thanking you in anticipation I am looking forward a favorable reply.

With Best Regards

Prof. O-Bong Yang

School of Semiconductor and Chemical Engineering

Jeonbuk National University, Jeonju 54896, Republic of Korea

Responses to Reviewer #1:

  1. Recommendation: Major revision

  2. Comments to Author:

Manuscript Number: applsci-901441

Title: Vertically Arranged Zinc oxide Nanorods as Antireflection layer for Crystalline Silicon Solar cell: A Simulation Study of Photovoltaic Properties

Overview and general recommendation:

This manuscript showed the antireflection layer of vertically arranged ZnO NRs on c-Si solar cells and studied the charge transport and photovoltaic properties by simulation. The authors optimized the reaction time for best surface and highly uniform ZnO NRs on the Si wafer. The optimum conversion efficiency of 14.88% has been achieved at 1.1 µm of ZnO NRs length with minimum average reflectance 9.5% by using this simulating tool.

However, the concept of this manuscript is similar to a previous report (doi:10.1016/j.solmat.2010.01.005). The previous report also used ZnO NRs as antireflection layer on silicon solar cells and showed that the efficiency was improved. When comparing the author's paper with the previous paper published in 2010, I could not find the difference and excellence. Also, there is a lack of explanation in this manuscript.

Answer: Thanks for your valuable suggestion. As compared to previous report (doi:10.1016/j.solmat.2010.01.005), the growth of vertically arranged ZnO AR on Si wafer is very easy, simple and highly economical. In this work, the growth process of NRs was carried out in airtight bottle which was placed in laboratory oven at low temperature of 75oC for 2-6 h. While, the growth temperature of ZnO NRs in reported is 100oC. Moreover, morphological features and crystalline properties of grown ZnO NRs are better than that of reported paper. Importantly, the reflectance of vertically arranged ZnO NRs as AR in solar cells exhibited much lower (average reflectance = 9.58%) to the reported paper (i.e. 20%). PC1D simulation results revealed that solar device with vertically arranged ZnO NRs as AR shows the superior photovoltaic properties such as PCE = 14.88%, Isc = 0.4970 A and Voc = 0.5894 V to reported one. This work is different from reported paper in terms of synthetic procedure, quality, structural and photovoltaic properties. 

Major recommendation:

Remark 1: The authors compared the different reaction time for 3, 4 and 5 hours. Among them, the performance of the simulation for 5 h ZnO NRs AR based Si solar cells shows the highest efficiency. The author is better to compare the 6 h ZnO NRs because the 5 h ZnO NRs lacks credibility to say that it is the optimal reaction time.

Reply:  Thanks for your valuable correction. As per reviewer’s suggestion, all the results of 6 h sample are included in the revised manuscript. FESEM, optical properties like reflectance, refractive index, extinction coefficient and photovoltaic parameters for 6 h sample have been included. The considerable decline in the all results including photovoltaic performance as compared to 5 h sample. The observation clearly reflects that 5 h ZnO NRs is the optimized condition to achieve the effective antireflection materials. 

Remark 2: In figure 5(a), please explain the reason for the shift of the wavelength and increase of the intensity by changing the reaction time.

Reply: Thanks for your valuable correction. PL measurement for 5 h sample was rechecked and found that no shifting occurred in the emission peaks. The new PL result is now included in the revised Fig. 5(a). 

Remark 3: Why doesn’t the intensity of the sample (b) 4h show a Si (100) peak?

Reply: Thanks for your valuable correction. We sincerely apology for using the wrong XRD data of 4 h sample. Now the Fig. 4 corrected and explained in revised manuscript.

Reviewer 2 Report

The work by Deb Kumar Shah, Devendra KC, M. Shaheer Akhtar, Chong Yeal Kim & O-Bong Yang reports antireflection layer of vertically arranged ZnO nanorods on crystalline silicon solar cells where they compare the charge transport and photovoltaic properties with those obtained from simulations. The study found that ZnO nano rods with antireflection layers have effective light absorption with better electrical performance. The authors further used various techniques to characterize the morphological and structural properties of the samples. While the goal of the study is clearly formulated, it is not very clear to me what is the novel in contrast to previous studies by others in the field. The previous studies (Zezeng Feng, et al Solar Energy 115: 770–776 (2015), Seong-Ho Baek, et al. J Mater Sci 47:4138–4145 (2012), Feifei Huang et al. J Mater Sci 54:4011–4023 (2019), Seong-Ho Baek, et al. Solar Energy Materials &Solar Cells 96:251–256 (2012)) discloses a silicon substrate enabled by controlled ZnO nanorods coating. In these experiments, the fabrication processes consisted of metal-assisted chemical seeding combined with coating and low temperature annealing for 1 h. In view of this, the work by Deb Kumar Shah and co-workers has already been covered explicitly in the previous studies.

Minor comments

  • For lines 52-54, the authors should consider revising to remove minor editorial errors and improve the clarity in sentence construction.
  • For lines 56-57, the authors should consider revising to remove minor editorial errors and improve the clarity in sentence construction.
  • For lines 59-62, the authors should consider revising to remove minor editorial errors and improve the clarity in sentence construction.
  • For lines 63-64, the authors should consider revising to remove minor editorial errors and improve the clarity in sentence construction.
  • For lines 67-71, the authors should consider revising to remove minor editorial errors and improve the clarity in sentence construction.
  • For lines 76-79, the authors should consider revising to remove minor editorial errors and improve the clarity in sentence construction.
  • For lines 79, it is not clear as what is so unique about vertically arranged nano rods.
  • For lines 114-115, the authors should consider revising to remove minor editorial errors and improve the clarity in sentence construction.

Author Response

Responses to Reviewers#2

Dated: August 20, 2020

Prof. Sadia Ameen

Guest Editor

Applied Sciences

Dear Editor

Thank you for forwarding the reviewer’s comments about our manuscript entitled “Vertically Arranged Zinc oxide Nanorods as Antireflection layer for Crystalline Silicon Solar cell: A Simulation Study of Photovoltaic Properties” with manuscript reference number “applsci-901441” by D.K. Shah et al.

Please find attached herewith the revised manuscript along with the reply sheet to honorable referees. You have kindly given us an opportunity for the revision after doing the needful modifications as per reviewer’s suggestions. Now, we have carefully revised and compensated all the reviewer’s comments in the revised manuscript. The corrected parts are highlighted by yellow background in the revised manuscript.

We hope that we have satisfactorily addressed all the valuable comments of the reviewers.

While thanking you in anticipation I am looking forward a favorable reply.

With Best Regards

Prof. O-Bong Yang

School of Semiconductor and Chemical Engineering

Jeonbuk National University, Jeonju 54896, Republic of Korea

Responses to Reviewer #2:

  1. Recommendation: Minor revision

  1. Comments to Author:

Manuscript Number: applsci-901441

Title: Vertically Arranged Zinc oxide Nanorods as Antireflection layer for Crystalline Silicon Solar cell: A Simulation Study of Photovoltaic Properties

Overview and general recommendation:

The work by Deb Kumar Shah, Devendra KC, M. Shaheer Akhtar, Chong Yeal Kim & O-Bong Yang reports antireflection layer of vertically arranged ZnO nanorods on crystalline silicon solar cells where they compare the charge transport and photovoltaic properties with those obtained from simulations. The study found that ZnO nano rods with antireflection layers have effective light absorption with better electrical performance. The authors further used various techniques to characterize the morphological and structural properties of the samples. While the goal of the study is clearly formulated, it is not very clear to me what is the novel in contrast to previous studies by others in the field. The previous studies (Zezeng Feng, et al Solar Energy 115: 770–776 (2015), Seong-Ho Baek, et al. J Mater Sci 47:4138–4145 (2012), Feifei Huang et al. J Mater Sci 54:4011–4023 (2019), Seong-Ho Baek, et al. Solar Energy Materials &Solar Cells 96:251–256 (2012)) discloses a silicon substrate enabled by controlled ZnO nanorods coating.

In these experiments, the fabrication processes consisted of metal-assisted chemical seeding combined with coating and low temperature annealing for 1 h. In view of this, the work by Deb Kumar Shah and co-workers has already been covered explicitly in the previous studies.

Answer: Thanks for your valuable suggestion. As compared to previous report (mentioned literature), the growth of vertically arranged ZnO AR on Si wafer is very easy, simple and highly economical. In this work, the growth process of NRs was carried out in airtight bottle which was placed in laboratory oven at low temperature of 75oC for 2-6 h. While, the growth temperature of ZnO NRs in reported is 100oC. Moreover, morphological features and crystalline properties of grown ZnO NRs are better than that of reported paper. Importantly, the reflectance of vertically arranged ZnO NRs as AR in solar cells exhibited much lower (average reflectance = 9.58%) to the reported paper (i.e. 20%). PC1D simulation results revealed that solar device with vertically arranged ZnO NRs as AR shows the superior photovoltaic properties such as PCE = 14.88%, Isc = 0.4970 A and Voc = 0.5894 V to reported one. This work is different from reported paper in terms of synthetic procedure, quality, structural and photovoltaic properties.

Minor recommendation:

Remark 1: For lines 52-54, the authors should consider revising to remove minor editorial errors and improve the clarity in sentence construction.

Reply: Thanks for your correction. The following sentence is now corrected in the revised manuscript as; “The ZnO NRs as AR layer considerably induces the reflection in a broad wavelength region at different incident angles because of continuous transition in the effective refractive index can be regarded across the air-to-wire interface.”

Remark 2: For lines 56-57, the authors should consider revising to remove minor editorial errors and improve the clarity in sentence construction.

Reply: Thanks for your correction. The following sentence is now corrected in the revised manuscript as; “The AR behavior of the ZnO NRs has owned a lot of attention because of its low dielectric constant and appropriate refractive index.”

Remark 3: For lines 59-62, the authors should consider revising to remove minor editorial errors and improve the clarity in sentence construction.

Reply: Thanks for your correction. The following sentence is now corrected in the revised manuscript as; “Including other deposition techniques for ZnO NRs, the low temperature solution method can offer a highly suitable method for controlling the NRs sizes and densities over the Si surfaces at low temperature.”

Remark 4: For lines 63-64, the authors should consider revising to remove minor editorial errors and improve the clarity in sentence construction.

Reply: Thanks for your correction. The following sentence is now corrected in the revised manuscript as; “The absorption of light or multiple reflection can be realized by deploying multiple transparent layers or zigzag structured on the top surface of C-Si solar cell.”

Remark 5: For lines 67-71, the authors should consider revising to remove minor editorial errors and improve the clarity in sentence construction.

Reply: Thanks for your correction. The following sentence is now corrected in the revised manuscript as; “SiC-SiO2 AR coated Si solar cell presented the similar PCE of 16.99% with SiNx AR coated Si solar cells”

Remark 6: For lines 76-79, the authors should consider revising to remove minor editorial errors and improve the clarity in sentence construction.

Reply: Thanks for your correction. The following sentence is now corrected in the revised manuscript as; “Therefore, the reflectivity highly decreased by growing of rod like nanostructures on the top surface of Si substrate and ZnO NRs can act as AR layer.”

Remark 7: For lines 79, it is not clear as what is so unique about vertically arranged nano rods.

Reply: Thanks for your correction. In order to avoid the confusion, the word ‘unique’ is omitted.

Remark 8: For lines 114-115, the authors should consider revising to remove minor editorial errors and improve the clarity in sentence construction.

Reply: Thanks for your correction. The following sentence is now corrected in the revised manuscript as; “In this work, the PC1D simulation has been performed using the length and average reflectance ………….”

Round 2

Reviewer 1 Report

The main concerns are cleared. So, it could be published in this journal.

Reviewer 2 Report

The changes by the authors are appreciated.